# Social Support and Its Impact on Job Satisfaction and Emotional Exhaustion

Pablo Garmendia [1], Samuel Fernández-Salinero [2,*], Ana Isabel Holgueras González [3] and Gabriela Topa [4]

1    Escuela Internacional de Doctorado de la UNED (EIDUNED), 28040 Madrid, Spain;
     pgarmendi4@alumno.uned.es
2    Psychology Department, Universidad Rey Juan Carlos, 28933 Madrid, Spain
3    Faculty of Education, National Distance Education University (UNED), 28040 Madrid, Spain;
     aiholgueras@edu.uned.es
4    Department of Social and Organizational Psychology, National Distance Education University (UNED),
     28040 Madrid, Spain; gtopa@psi.uned.es
*    Correspondence: samuel.fernandez@urjc.es

**Abstract:** Social support at work has demonstrated itself to be an important variable for predicting desirable outcomes and helping to buffer the effects of adverse events. The main objective of this research is to understand the impact of social support on job satisfaction on the one hand and emotional exhaustion on the other. Furthermore, in order to gain a deeper understanding of intricate organizational relationships, the mediating effects of work recovery experiences are taken into consideration. The sample was composed of 496 workers (41.5% men and 58.5% women). The mean age was 42 years (SD = 9.82). A cross-sectional design was used. The results, both direct (r = 0.43; $R^2$ = 0.19; $p < 0.001$) and indirect (B = 0.04; SE = 0.02; 95% C.I. = 0.01, 0.09), of the model relating social support to job satisfaction were statistically significant. On the other hand, in the model that links social support to emotional exhaustion, we observed statistically significant direct (r = 0.26; $R^2$ = 0.07; $p < 0.001$) and indirect effects (B = −0.05; SE = 0.02; 95% C.I. = −0.10, −0.01). Only the relaxation factor was a significant mediator of these variables. Implications, limitations, and future research recommendations are discussed.

**Keywords:** social support; job satisfaction; emotional exhaustion; recovery experiences

## 1. Introduction

Current organizations are concerned with promoting health among their employees. Corporate social responsibility is an interesting topic that organizations are embracing to foster their own benefits and those of their employees. DiFabio [1] emphasizes that the 21st century is characterized by complexity and that well-being in organizations is key to the development of health, characterized not only by the absence of illness but also as a state of complete physical, mental, spiritual, and social well-being. [2] Furthermore, the emergence of Positive Psychology has brought about a shift towards well-being based on the enhancement of organizational and personal resources [1,3]. Specifically, there has been a transition from the traditional view focused on eliminating risk factors to an emphasis on growth and the promotion of positive experiences.

It is crucial to emphasize at this point that work not only serves survival needs but also relates to the need for social connection [1]. In other words, there is a need for relationships. Therefore, labor relationships are of paramount importance in organizations and are approached from the perspective of sustainable organizations. Social relationships, when they are constructive, contribute to improving the quality of life and depend on the significance that individuals and the organization attach to them, as well as their importance in the specific context of the workers [1]. For this reason, recent research has underscored the necessity of recognizing and respecting the importance of social

relationships in organizational contexts to promote healthier environments. In this line, it has been widely proven that social support is related to desirable health outcomes such as job satisfaction [4] or helps to buffer undesirable outcomes such as emotional exhaustion [5]. A lot of recent research has been conducted on nurses [5] or in the academic field [6], but it is important to understand how these relationships occur among general workers. Furthermore, in the academic literature on sustainability, the importance of social aspects has been recognized by introducing concepts such as social capital [6,7], which are highly relevant and worth considering.

However, the underlying mechanisms of how social support can operate remain a mystery. Therefore, this research is based on this assumption and seeks to contribute to the scientific literature by providing a deeper understanding of the modulations that occur between social support and organizational health variables. Within this context, job satisfaction is an important variable to consider. Job satisfaction is associated with an individual's perception of their work. Some authors have linked it to the affective component of attitudes towards one's job. The interaction component has consistently been related to current job satisfaction in classical research [8] and has been emphasized as an important component in subsequent studies [9]. Additionally, our research is grounded in the suggestions of Niskala et al. [10], who propose in their meta-analysis that understanding the factors affecting job satisfaction is of great importance for effective organizational management. One of the justifications for this study is to verify through which individual processes social support enhances job satisfaction.

In this line, we consider that recovery from experiences has been stated as a key variable in guaranteeing vigor, motivation, and health among employees, even when facing stressful situations [11]. Not only off-the-job recovery processes but also in-the-job ones are important to cope with daily stressors. Moreover, perceived recovery is more important than physical recovery. Nowadays, work occupies most of an individual's time and energy compared to previous generations [12]. This circumstance may make it difficult for the daily options to disconnect and recover. However, the key to recovery is that individuals may recover from work stressors even during small breaks during the workday [12,13]. Recovery may occur even within one's job. On the other hand, it is necessary to explore deeper the relationships between social support and recovery experiences. This article aims to understand if there is a relationship between these variables and whether it is possible that a mediation mechanism exists to help understand the conditions under which both burnout and job satisfaction occur. This is where our research provides a novel perspective. Previous meta-analysis has stated that recovery must be precisely conceptualized, while previous research has focused the majority on off-job activities [12,14]. Steed et al. [12] stated that recovery has been conceptualized differently in previous literature (activities, experiences, states, etc.). The new trends in sustainability literature refer to not only avoiding hindering the processes that occur within organizations but also paying attention to those processes that can contribute to regenerating and renewing resources [15]. Given the fact that recovery is a relatively new concept, mixed empirical outcomes have been identified, and there is a need to understand which factors are important in predicting health variables. Therefore, since there is a research gap in understanding the relationships between social support and recovery, one of the main contributions of this paper is to try to fill this gap.

Finally, one of the variables that has received considerable attention in the organizational context due to its undesirable outcomes is burnout. Burnout is conceptualized as a psychosocial syndrome that occurs as a reaction to chronic stress exposure [16]. Burnout consists of three factors: emotional exhaustion, cynicism, and reduced personal accomplishment. One of the most critical components of burnout, which has been associated with a strong impact on other organizational variables, is emotional exhaustion. Emotional exhaustion is related to feelings of being overwhelmed and drained due to work. The selection of emotional exhaustion as relevant to this research is based on the systematic review conducted by XX with teachers, which asserted that affective symptoms were related to the

feeling of lacking the energy to cope with work demands [17]. Building upon this reflection, this research seeks to shed light on the question of whether recovery processes that may be activated through social support can help reduce the sense of exhaustion and lack of energy in workers.

Having said this, this research focuses on trying to clear the relationships between social support and, on the one hand, a desirable job outcome (job satisfaction) and, on the other hand, an undesirable job outcome (emotional exhaustion). Since the organizational field is complex, recovery experience factors are intended as mediators of these relationships. The main objective of this research is to explore and extend the previous research findings between social support, job satisfaction, and emotional exhaustion, using recovery experiences as mediators.

## 2. Theoretical Framework

Social Support: Social support refers to social and psychological support that an individual receives or perceives from its environment [18]. As previous research suggests, there are two kinds of social support [6]. On the one hand, received social support refers to the degree to which a person has received support from their environment, while perceived social support refers to the perception and availability of this support [19].

The term 'social support' has been defined in various ways within the academic field of psychology. Cohen & Syme [20] provided a broad definition, describing it as all resources provided by others. Furthermore, social support has been defined in numerous studies as a comprehensive term encompassing all aspects of social interactions [21]. Due to these conceptual challenges, alternative approaches have sought to adopt a more outcome-oriented perspective, defining social support based on the results it yields.

Subsequently, in an effort to structurally operationalize social support, various categories were established to facilitate a scientific approach to the term. Following Schwarzer & Leppin [21], these proposed dimensions included structural versus functional, received versus perceived, global versus specific, presence of versus satisfaction with support, and given versus received. Regarding the outcomes, research has produced remarkable findings indicating that various types of social support influence both physical and mental health variables. Some studies have even linked social support to diseases such as cancer and coronary heart disease.

This variable has proven to be a very important factor in the organizational field. It has been not only related to positive individual outcomes such as positive affect [22], but also to a buffer against stress. As previous literature states, social support aids individuals to cope with problems, improving positive psychological and behavioral responses [6]. Moreover, recent research has shown that social support programs in the workplace are related to improvements in employees' well-being. On the other hand, some scholars have shown the direct and indirect effects of occupational stress and overemotional exhaustion [23]. Therefore, its influence may be double. First, social support may enhance job satisfaction (Hypothesis 1) and, complementarily, it may help to lower exhaustion levels (Hypothesis 2).

In our research, aiming to gain a deeper understanding of the organizational relationships established between the variables, we have endeavored to comprehend how social support can impact variables such as job satisfaction and emotional exhaustion. Specifically, we seek to test the hypothesis that social support may correlate with these variables through the factors of work recovery experiences that it may facilitate.

Job satisfaction: Job satisfaction involves emotional parameters related to one's work, such as pleasure, happiness, passion, or enthusiasm, that an individual feels towards their dedication to their job [24]. It is associated with experiences of contentment within the workplace. Previous research has attempted to provide more concise definitions of the construct, and some have proposed it as positive affect derived from work experiences. Job satisfaction has demonstrated correlations with significant organizational variables, such as physical and mental health, and even talent retention within organizations [24].

It has been posited that job satisfaction may be influenced by a variety of factors, such as the organizational culture, the management style, or the relationships with coworkers [25]. Social support has been demonstrated to be a critical factor in predicting job satisfaction [26]. On the other hand, intraorganizational communication has been related to job satisfaction by recent scholars [27]. Since job satisfaction is an important and multidimensional factor, understanding the ways in which social support enhances job satisfaction helps foster healthy organizations. Given that job satisfaction is a measure stemming from the experience of positive events within the workplace, this research aims to shed light on whether the recovery experience resulting from social support can influence job satisfaction. Additionally, we aim to identify which recovery factors can exert a more decisive impact on satisfaction. Therefore, we intend to test the hypothesis that social support will correlate with job satisfaction, and furthermore, we believe it may do so through the facilitated recovery experience.

In addition, previous investigations have shown that social support tends to decrease emotional exhaustion [5,6]. Emotional exhaustion is the primary component of burnout, and it is related to the subjective feeling of being emotionally exhausted and depleted of emotional resources [28]. Emotional exhaustion is usually related to occupations with a high level of interpersonal interaction [29,30]. The relationship between social support and the decrease of emotional exhaustion has been observed in most of the results of the Velando-Soriano et al. [5] systematic review. It has been stated that emotional exhaustion leads to negative attitudinal, emotional, and behavioral outcomes. For example, exhaustion has a negative effect on job satisfaction, organizational commitment [31], performance [32], and turnover intentions [33]. Ariapooran [34] emphasized the central role of social support in buffering emotional exhaustion. In our research, considering, as mentioned earlier, that both social support and recovery experiences can be affective factors that enhance the work experience, we intend to test the hypothesis of whether there is a reduction in emotional exhaustion. Specifically, our goal is to assess whether social support can lead to a work recovery experience and, through this interaction, promote a decrease in an individual's levels of emotional exhaustion.

Recovery: Some previous scholars have stated that the opportunity to recover from demanding situations is a key factor in avoiding the negative consequences for health and well-being [35,36]. Recovery has been defined as the experience of psychophysiological relaxation after facing a stressful event [11]. So, recovery is seen as the procedure through which a person stops coping with a demanding situation to renew its resources. Initial research on this topic showed that recovery was directly related to well-being and performance [37]. Some interesting insights that the authors purposed about this variable are the fact that recovery is more a psychological experience than a physical process. It is well known that recovery experiences are related to experiences outside the workplace. In this line, some scholars have related social support outside the workplace as an important variable predicting self-blame and food or substance use as coping strategies [38]. But recovery may occur not only as an off-job experience but also in the workplace.

Recovery is a variable comprised of four factors. Psychological detachment, relaxation, mastery, and control. The first two factors are derived from the Effort-Recovery Model, while the last two are derived from the Conservation of resources model [39]. Psychological detachment is related to the capacity to mentally disengage during off-the-job time. Relaxation is a state characterized by low activation and an increasing positive effect. On the other hand, Mastery is related to the capability of growth in off-the-job activities that offer new challenges and opportunities. This mastery is related to skill development, competence, self-efficacy, and a positive mood [40]. Last, control is related to the ability to decide when and how to pursue some leisure activities.

While social support has been demonstrated to be an interesting factor related to emotional regulation processes [41], relaxation may occur even in the workplace. Therefore, based on the previous literature and to complement the main hypotheses of this work, we propose the following mediation hypotheses: As previously mentioned, given that

recovery experiences are associated with higher levels of satisfaction and lower levels of emotional exhaustion, we can hypothesize that social support might trigger these recovery processes within the workplace. Thus, the mediation hypotheses are proposed in Table 1 as sub-hypotheses of Hypotheses 1 and 2.

**Table 1.** List of hypotheses and descriptions.

| Hypotheses | Description |
| --- | --- |
| H1 | Social support will be statistically significant and directly correlated with job satisfaction. |
| H1a | Psychological detachment will mediate the relationships between social support and job satisfaction. |
| H1b | Relaxation will mediate the relationships between social support and job satisfaction. |
| H1c | Mastery will mediate the relationships between social support and job satisfaction. |
| H1d | Control will mediate the relationships between social support and job satisfaction. |
| H2 | Social support will be statistically significant and inversely correlated with emotional exhaustion. |
| H2a | Psychological detachment will mediate the relationships between social support and emotional exhaustion. |
| H2b | Relaxation will mediate the relationships between social support and emotional exhaustion. |
| H2c | Mastery will mediate the relationships between social support and emotional exhaustion. |
| H2d | Control will mediate the relationships between social support and emotional exhaustion. |

Social support may act differently than emotional exhaustion and job satisfaction, and the main objective of this paper is to understand how the positive impact of social support occurs. Since recent research has highlighted the importance of detachment in physiological variables such as sleep quality [42], understanding which factors of recovery experience help to enhance health is another aim of this research. In addition, there is a lack of research relating social support to recovery at work. Most of the studies relate social support to other important health outcomes, such as recovery after an injury [43], mental health [44–46], or recovery from substance use [47]. Since previous research has shown that social support is a very strong predictor of health outcomes, studying the relationships between it and recovery experiences at work is relevant. Moreover, because recovery factors have not been posed as mediators between social support and the criterion variables, this research may help to gain a deeper understanding of this phenomenon.

Having said this, we summarize our working hypotheses in Table 1.

## 3. Materials and Methods

### 3.1. Participants

The sample was composed of 496 subjects (41.5% men and 58.5% women). The mean age was 42 years (SD = 9.82). Regarding tenure, the mean age in the same organization was 13 years (SD = 9.95). The median was 11. In addition, we considered the education degree of our sample. A total of 60.9% of the sample had a university degree, 24.2% had vocational training, 7.9% had a bachelor's degree, and 7.1% had completed primary education. Related to the professional category, 5.6% of our sample were managers, while 12.1% were middle-level managers. Moreover, 7.3% were administrative employees, and 19.6% were technical employees. Lastly, 46.4% were qualified employees, and the remaining 9.1% were non-qualified employees. Educational level completed, professional category, tenure, and professional sector can be referenced in detail in Table 2.

**Table 2.** Descriptive statistics regarding the sample of the current study.

| | Men *n* (%) | Women *n* (%) | Total *n* (%) |
| --- | --- | --- | --- |
| Total | 206 (41.5%) | 290 (58.5%) | 496 (100%) |
| Educational level | | | |
| University Degree | 109 (52.9%) | 193 (66.6%) | 302 (60.9%) |
| Vocational training | 58 (28.2%) | 62 (21.4%) | 120 (24.2%) |
| Bachelor | 20 (9.7%) | 19 (6.6%) | 39 (7.9%) |
| Primary education | 19 (9.2%) | 16 (5.5%) | 35 (7.1%) |

**Table 2.** *Cont.*

|  | Men *n* (%) | Women *n* (%) | Total *n* (%) |
|---|---|---|---|
| **Professional category** | | | |
| Manager | 18 (8.7%) | 10 (3.4%) | 28 (5.6%) |
| Middle-level manager | 33 (16%) | 27 (9.3%) | 60 (12.1%) |
| Administrative employees | 14 (6.8%) | 22 (7.6%) | 36 (7.3%) |
| Technical employees | 43 (20.9%) | 54 18.6%) | 97 (19.6%) |
| Qualified employees | 77 (37.4%) | 153 (52.8%) | 230 (46.4%) |
| Non-qualified employees | 21 (10.2%) | 24 (8.3%) | 45 (9.1%) |
| **Job Tenure** | | | |
| 0–5 years | 51 (24.8%) | 83 (28.6%) | 134 (27.0%) |
| 6–10 years | 55 (26.7%) | 61 (21.0%) | 116 (23.4%) |
| 11–15 years | 27 (13.1%) | 41 (14.1%) | 68 (13.7%) |
| 16–20 years | 22 (10.7%) | 48 (16.6%) | 70 (14.1%) |
| 21–25 years | 21 (10.2%) | 24 (8.3%) | 45 (9.1%) |
| +25 years | 30 (14.5%) | 33 (11.4%) | 63 (12.7%) |
| **Professional sector** | | | |
| Primary | 3 (1.5%) | 4 (1.4%) | 7 (1.5%) |
| Secondary | 67 (32.5%) | 23 (7.9%) | 90 (18.1%) |
| Tertiary | 136 (66.0%) | 263 (90.7%) | 399 (80.4%) |

*3.2. Procedure*

To conduct the present research, several Spanish organizations were contacted via email and asked to voluntarily participate. We request permission to distribute the form among the workers. This study was conducted in accordance with the Declaration of Helsinki for studies involving humans and authorized by the ethics committee of the National Distance Education University. The sampling procedure was carried out for convenience. The questionnaire was filled out via the Google Forms application. Participants were informed about the anonymity and confidentiality of their contributions. Moreover, participants were informed about voluntary abandonment and the possibility of abandonment. There were no exclusion criteria for participating, and as we shared the link to the research, it was not possible to collect data about the participation rate. All the analyses were conducted using IBM SPSS v.24. For testing mediation models, we used Model 4 for the PROCESS macro, developed by Hayes [48]. Firstly, the data were downloaded from the Google Forms platform, and it was ensured that no data were missing by setting all questions as mandatory. Subsequently, after sorting the variables and calculating the questionnaire means, descriptive statistics, frequency tables, and cross-tabulations were computed. Finally, the necessary assumptions for regression models (linearity, homoscedasticity, and normal distribution of the error) were tested. Once these assumptions were verified, the PROCESS macro by Hayes [48] was used to test the parallel mediation model. The PROCESS macro is a plugin for the IBM SPSS software that provides a graphical interface for testing various statistical regression models. The PROCESS macro is a statistical tool that enables these analyses. In general terms, the macro facilitates the execution of multiple regression models to evaluate mediation and moderation relationships in the data. It calculates regression coefficients and their standard errors to assess whether the assumptions of mediation are met. The Hayes macro assists in implementing these analyses and provides detailed results that help understand the strength and direction of relationships between variables, as well as the presence or absence of mediation. It is a useful tool for researchers aiming to explore and better understand the dynamics between variables in their studies. Specifically, the parallel mediation model is Model 6. The program generates 1000 random samples from the dataset to perform hypothesis testing. This technique helps researchers understand how and to what extent a mediator variable explains the relationship between an independent and a dependent variable in a statistical model.

### 3.3. Measures

Social Support: To assess social support, we used the Spanish version [49] of the work design questionnaire [50]. Specifically, we used the social support subscale, which is comprised of a 6-item Likert-type scale. The scale is rated from 1 (strongly disagree) to 5 (strongly agree). Examples of the items are "I have the opportunity to develop close friendships in my job" or "People I work with take a personal interest in me." The validation of the scale showed a reliability index of $\alpha = 0.80$, while in our sample, the social support factor showed $\alpha = 0.82$.

Recovery: For assessing recovery experiences, we used the Spanish validation [36] of the Recovery Experience Questionnaire [41]. This questionnaire is composed of a 5-point Likert-type scale. The scale ranges from 1 (totally disagree) to 5 (totally agree). It consists of 16 items divided into 4 factors (Psychological detachment, Relaxation, Mastery, and Control). Each factor consists of four items. Examples of the items are "I distance myself from my work" (psychological detachment), "I do relaxing things" (Relaxation), "I do things that challenge me" (Mastery) and "I decide my own schedule" (Control). Related to reliability, the validated scale showed an $\alpha = 0.82$ for psychological detachment, $\alpha = 0.74$ for relaxation, $\alpha = 0.84$ for Mastery, and $\alpha = 0.87$ for control. In our sample, reliability indexes were $\alpha = 0.86$, $\alpha = 0.84$, $\alpha = 0.87$, and $\alpha = 0.88$, respectively.

Job Satisfaction: In the case of job satisfaction, the Spanish validation [51] of the Brief Index of Affective Job Satisfaction (BIAJS) [52] was used. This questionnaire is composed of four items. The Likert-type scale ranged from 1 (strongly disagree) to 5 (strongly agree). The scale only satisfies one Affective Job Satisfaction factor. Anyway, the scale includes three distracting items. Examples of these items are "I feel fairly well satisfied with my job" and "I like my job better than the average person." Related to reliability index, the original validation showed a Cronbach's alpha of $\alpha = 0.83$, while in our sample, the construct showed a reliability of $\alpha = 0.91$.

Emotional exhaustion. Lastly, emotional exhaustion was measured with one factor of the Spanish validation [53] of the Maslach Burnout Inventory [54]. Specifically, this factor is composed of 5 items on a 7-point Likert-type scale ranging from 0 (never) to 6 (every day). Examples of these items are "I feel emotionally exhausted because of my work" or "I feel worn out at the end of a working day." The reliability index of the original work was $\alpha = 0.85$, and in our sample, the index was $\alpha = 0.90$.

## 4. Results

### 4.1. Descriptive an Correlations

The first step of our study was to check the descriptive statistics of our variables and verify their correlations. Table 3 displays the Pearson correlation coefficients among the different variables included in this study.

**Table 3.** Means, standard deviations, and correlations between the variables.

|  | M | M | SD | $\alpha$ | 1 | 2 | 3 | 4 | 5 | 6 | 7 | 8 | 9 |
|---|---|---|---|---|---|---|---|---|---|---|---|---|---|
| 1. Age | 42 | 9.82 | - | - | - | - | - | - | - | - | - | - | - |
| 2. Tenure | 13 | 9.95 | - | 0.75 ** | - | - | - | - | - | - | - | - | - |
| 3. Social Support | 3.84 | 0.63 | 0.82 | −0.19 ** | −0.16 ** | - | - | - | - | - | - | - | - |
| 4. Detachment | 3.28 | 0.82 | 0.86 | −0.09 * | −0.00 | 0.88 | - | - | - | - | - | - | - |
| 5. Relax | 3.51 | 0.76 | 0.84 | −0.11 * | −0.06 | 0.19 ** | 0.59 ** | - | - | - | - | - | - |
| 6. Mastery | 3.69 | 0.73 | 0.87 | −0.05 | −0.08 | 0.14 ** | 0.18 ** | 0.40 ** | - | - | - | - | - |
| 7. Control | 3.87 | 0.73 | 0.88 | 0.00 | 0.03 | 0.15 ** | 0.24 ** | 0.39 ** | 0.33 ** | | - | - | - |
| 8. Job Satisfaction | 3.50 | 0.85 | 0.91 | −0.02 | −0.02 | 0.43 ** | 0.01 | 0.19 ** | 0.17 ** | 0.12 ** | - | - | - |
| 9. Exhaustion | 2.42 | 0.83 | 0.90 | 0.10 * | 0.08 | −0.25 ** | −0.26 ** | −0.29 ** | −0.10 * | −0.11 * | −0.38 ** | - | - |

Note. N = 496; M = Mean; SD = Standard Deviation. ** $p < 0.01$; * $p < 0.05$.

After checking the data in Table 3 and considering the correlation coefficients, we can assert that we have significant evidence to support the correlation proposed in Hypothesis 1. As Table 2 shows, we have enough evidence to confirm Hypotheses 1 (r = 0.43; $p < 0.001$) and 2 (r = −25; $p < 0.001$). Having verified the relationship between these variables and reviewed the scatter plots, we can establish that the first assumption for conducting a regression model is met. To gain a deeper comprehension of how these relationships occur, we are conducting a parallel mediation model.

Since our sample showed that not only age but also tenure was statistically significantly related to other variables, we decided to include them as covariates in our mediation models.

### 4.2. Mediation

To test sub-hypotheses 1 and 2, we developed a parallel mediation model based on Hayes' PROCESS macro for IBM SPSS v.24 [48]. Specifically, model 4 was tested with 1000 samples using the bootstrapping procedure. The indicators considered to assess the significance of the regression models are the regression beta coefficients, which indicate the strength and direction of relationships. Additionally, confidence intervals are taken into account, and if they do not include the value 0, it indicates that the effect is statistically significant. Furthermore, the determination coefficients inform us about the proportion of variability explained by the model, and finally, they evaluate whether the indirect effect is statistically significant. Therefore, the program obtains 1000 random samples from the original database and tests the models to subsequently generate evaluation indicators. For a clearer exposition, we will first show the results of the job satisfaction model and then the emotional exhaustion model. Direct effects refer to the total impact of the independent variable on the dependent one without considering the mediator's influence. On the other hand, indirect effects take into account the mediator's interaction. The indirect effect is composed, on the one hand, of the effect of the independent variable on the mediator and, on the other hand, of the mediator's effect on the dependent variable. Lastly, the total effect comprises both the direct and indirect effects. Considering these indices helps in understanding how independent and mediator variables influence the dependent variable.

Social support for Job Satisfaction Direct effects. We found that the regression model of social support on job satisfaction was statistically significant (r = 0.43; R2 = 0.19; $p < 0.001$). Specifically, including the mediator variables, the total effect was significant (B = 0.60; SE = 0.06; 95% C.I. = 0.49, 0.71), while the direct effect model was significant as well (B = 0.56; SE = 0.06; 95% C.I. = 0.45, 0.68). Based on this data, we can affirm that social support is clearly related to job satisfaction.

Social support for Job Satisfaction Indirect effects. The indirect effects model was significant in our sample (B = 0.04; SE = 0.02; 95% C.I. = 0.01, 0.09). Specifically, we found that only relaxation showed a significant effect as a mediator (B = 0.04; SE = 0.02; 95% C.I. = 0.01; 0.09). Hence, we have enough evidence to confirm Hypothesis 1b. Nor psychological detachment (B = −0.01; SE = 0.01; 95% C.I. = −0.04; 0.01) (H1a), mastery (H1c) (B = 0.01; SE = 0.01; 95% C.I. = −0.00; 0.04) or control (H1d) (B = 0.00; SE = 0.01; 95% C.I. = −0.02; 0.02) showed a significant interaction. Age (B = 0.01; SE = 0.01; $p > 0.05$) and tenure (B = 0.00; SE = 0.01; $p > 0.05$) did not show a statistically significant impact on the job satisfaction regression model. Due to this, sub-Hypothesis 1 may only be partially confirmed. Detailed data may be seen in Table 4 and Figure 1.

Social support for emotional exhaustion Direct effects. In the case of emotional exhaustion, we found that the regression model was statistically significant (r = 0.26; R2 = 0.07; $p < 0.001$). Specifically, including the mediator variables, the total effects were significant (B = −0.31; SE = 0.06; 95% C.I. = −0.43, −0.20), while the direct effect model was significant as well (B = −0.27; SE = 0.06; 95% C.I. = −0.39, −0.16). We can state that social support negatively impacts emotional exhaustion.

**Table 4.** Mediation model between social support and job satisfaction detailed results.

| Model | B | SE | LLCI | ULCI |
|---|---|---|---|---|
| Total model | 0.60 | 0.06 | 0.49 | 0.71 |
| Direct effects | 0.56 | 0.06 | 0.45 | 0.68 |
| Indirect effects | 0.04 | 0.02 | 0.01 | 0.09 |
| S.Sup. > Relax > J.Sat. | 0.04 | 0.02 | 0.01 | 0.09 |
| S.Sup. > Detachment > J.Sat. | −0.01 | 0.01 | −0.04 | 0.01 |
| S.Sup. > Mastery > J.Sat. | 0.01 | 0.01 | −0.00 | 0.04 |
| S.Sup. > Control > J.Sat. | 0.00 | 0.01 | −0.02 | 0.02 |

Note. S.Sup = Social Support; J.Sat = Job Satisfaction; B = Beta; SE = Standard error; LLCI = Lower limits confidence interval; ULCI = Upper limits confidence interval.

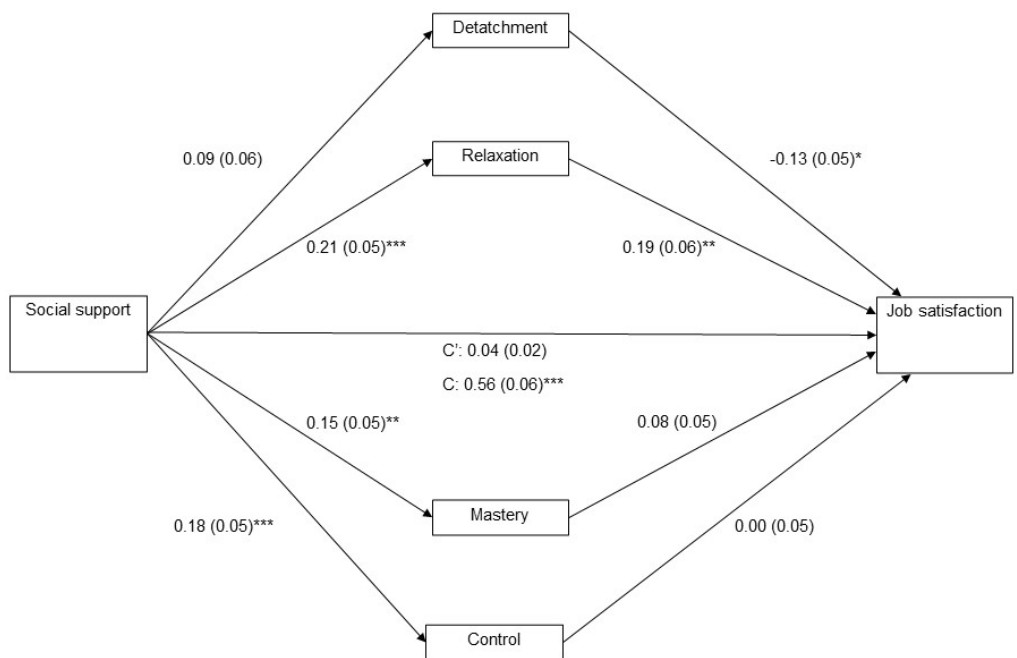

**Figure 1.** Parallel mediation model linking social support and job satisfaction (*n* = 496). Note: *** *p* < 0.001, ** *p* < 0.01; * *p* < 0.05. C = Direct effect of social support on job satisfaction; C′ = Indirect effect of social support on job satisfaction.

Social support for emotional exhaustion Indirect effects. In this case, we tested the indirect effects model. We found that it was significant in our sample (B = −0.05; SE = 0.02; 95% C.I. = −0.10, −0.01). Like in the previous section, we found that only relaxation showed a significant effect as a mediator (B = −0.04; SE = 0.02; 95% C.I. = −0.09; −0.01). With this data, we can confirm Hypothesis 2b. Psychological detachment (B = −0.01; SE = 0.01; 95% C.I. = −0.04; 0.00) (H2a), mastery (B = 0.00; SE = 0.01; 95% C.I. = −0.01; 0.03) (H2c), and control (B = 0.00; SE = 0.01; 95% C.I. = −0.02; 0.02) (H2d) did not show a significant impact. Anyway, age (B = −0.00; SE = 0.00; *p* > 0.05) and tenure (B = 0.00; SE = 0.01; *p* > 0.05) did not show any statistically significant impact in the general regression model. Hence, sub-hypothesis 2 may only be partially confirmed. Detailed results may be seen in Table 5.

**Table 5.** Mediation model between social support and emotional exhaustion detailed results.

| Model | B | SE | LLCI | ULCI |
|---|---|---|---|---|
| Total model | −0.31 | 0.06 | −0.43 | −0.20 |
| Direct effects | −0.27 | 0.06 | −0.39 | −0.16 |
| Indirect effects | −0.05 | 0.02 | −0.10 | −0.01 |
| S.Sup. > Relax > Exhaustion | −0.04 | 0.02 | −0.09 | −0.01 |
| S.Sup. > Detachment > Exhaustion | −0.01 | 0.01 | −0.04 | 0.00 |
| S.Sup. > Mastery > Exhaustion. | 0.00 | 0.01 | −0.01 | 0.03 |
| S.Sup. > Control > Exhaustion | 0.00 | 0.01 | −0.02 | 0.02 |

Note: S.Sup= Social Support; B = Beta; SE = Standard error; LLCI = Lower limits confidence interval; ULCI = Upper limits confidence interval.

## 5. Discussion

The main objective of this research was to explore and extend the previous research findings on social support, job satisfaction, and emotional exhaustion in a general worker sample. First, we can affirm that our research is on the line of previous research that relates social support with job satisfaction [6] on the one hand and with lower levels of emotional exhaustion on the other [5]. The primary contribution of our work is to understand the mechanisms through which the relationships found in previous research occur. In other words, we aim to understand the role of work recovery experiences in occupational health. Our research has contributed to the understanding that work-related relaxation facilitated by social support is an intriguing factor for both improving job satisfaction and reducing emotional exhaustion. It is necessary to expand the research that links social support and the experience of recovery within the work environment. Therefore, our article can shed light on this phenomenon and help understand this complex and novel reality. Future research should expand on these findings and confirm whether they occur in various populations and job settings. Additionally, exploring if the different components of work recovery experiences behave differently in various samples can be an interesting avenue of study.

Moreover, work recovery experiences have been associated with significant consequences both within and outside the work sphere. Specifically, recent studies have shown that work recovery is related to desirable outcomes such as engagement, life satisfaction, or a reduction in emotional exhaustion [55]. However, our research proposes work recovery as a mediating variable, and this may influence the results of recovery-related factors. Future research should investigate whether different components of recovery experiences behave differently when recovery occurs outside the work environment. It is possible that in recovery experiences outside of work, variables such as psychological detachment or mastery may behave differently in predicting satisfaction in other aspects of life.

In this vein, literature has supported the idea that relaxation and detachment can help cushion the effects of work stressors [56]. We propose the opposite phenomenon: the positive experience of social support is associated with greater relaxation, which may result in a decrease in emotional exhaustion and an increase in job satisfaction. Considering this approach is crucial for fostering healthy and sustainable organizations. Additionally, we also want to emphasize that work recovery within the job itself can contribute to broadening the perspective and design of job roles. It may be more challenging to achieve psychological distance within the job itself, and social support may not provide experiences of mastery in new skills or control over one's own time. Therefore, future research will need to verify if these same relationships occur in informal groups or outside the work environment. Therefore, the present research highlights that promoting social support can foster healthy dynamics among employees. This aligns with previous studies [57] that emphasized the importance of positive relational management [58]. This construct refers to a relational style characterized by respect and concern for oneself and others, as well as interpersonal sensitivity. In our research, we have taken a step forward to understand through which components of work recovery both the increase in satisfaction and the decrease in emotional exhaustion occur.

Following the proposals of DiFabio & Rosen [59], healthy organizations are characterized by seeking new solutions and generating resources that lead to a shift from a threat perspective to an opportunity perspective. In this regard, the importance of interpersonal connections has been emphasized [58], along with the possibility of training and intervention for managers to develop these resources. Due to this, current organizations should take these results into account and promote positive and desirable social support in order to trigger positive dynamics such as relaxation, which can lead to healthier outcomes. In this line, it seems that relaxation is the most important factor in work-recovery experiences, through which these relationships are established. This is on the line of previous research that confirmed the positive effects of relaxation in reducing anxiety levels or decreasing tension [60]. Social contact should aid employees to relax and feel comfortable even in the workplace, allowing them to feel more satisfied with their jobs and less exhausted. This is aligned with recent research that emphasizes the need to focus on renewable resources as well as the purification and oxygenation of processes within the work environment [15].

### 5.1. Implications for Occupational Health Practice

As previous research suggested, recovery may happen even inside one's job. Social support is correlated with every factor of recovery experiences but psychological detachment. This may occur because psychological detachment is more likely to happen in off-the-job activities where it has been reported to have desirable outcomes such as work engagement and lower levels of emotional exhaustion [61]. In our research, almost every factor of recovery experience correlates with both job satisfaction and emotional exhaustion. Future research should shed light on the mechanisms underlying these relationships. Only detachment did not show statistically significant relationships with job satisfaction. This is on the line of previous research, which relates job satisfaction to other direct sources such as the physical environment, relationships with colleagues, social support, and the quality of leadership [61,62]. Even when recovery experiences are related to criterion variables, future research should extend the evidence about the paths through which these relationships are established.

### 5.2. Applying Research to Occupational Health Practice

Social support has been confirmed to be a very important variable in enhancing job satisfaction and lowering levels of emotional exhaustion. These relationships were statistically significant. Organizations should focus on these variables in order to enhance the workplace. On the other hand, recovery factors (psychological detachment, relaxation, mastery, and control) were statistically significant and correlated with criterion variables. Only detachment did not correlate with job satisfaction. Current organizations should provide opportunities for employees to relax to foster healthier outcomes and sustainable workplaces. This study confirms that social support is an essential tool for implementing organizational sustainability policies and fostering the development of healthy organizations. Peer support plays a crucial role in promoting sustainable practices and creating a supportive work environment. By encouraging collaboration, knowledge sharing, and mutual assistance among colleagues, organizations can enhance their sustainability efforts and improve overall well-being. Since our data supports the idea that social support is directly related to job satisfaction through relaxation, managers should consider the chance to stimulate human interactions in the workplace. In addition, social support was inversely statistically significant and related to emotional exhaustion, which may be related to the use of healthier coping strategies.

### 5.3. Study Strenghts and Limitations

Our research possesses several strengths. Firstly, it contributes to understanding the dynamics of the relationship between social support and job satisfaction, on the one hand, and emotional exhaustion, on the other. The primary contribution of this article lies in the examination of the mediating relationships established through work recovery experiences.

This is a highly intriguing area that can aid in the development of healthier organizations. Our research has some limitations that must be considered. First, the sampling procedure is not randomized. This may imply representativity problems. Future research should use randomized procedures to avoid sampling biases. In addition, our sample has some sociodemographic idiosyncrasies. It is composed fairly by university graduates. Moreover, we did not collect professional categories. Future research should replicate these findings, collecting data for different job types and qualifications. Furthermore, we lack information concerning the integrity of the sample, which could potentially introduce bias into the data we have collected. Future research endeavors should assess whether a relationship exists between the variables under examination and the motivation to respond to questionnaires. In addition, it would be interesting to consider the values of perceived stress to evaluate if social support works differently among different levels of work-related stress. Another limitation is the fact that our research is a transversal study, which makes it impossible to establish causal relationships. Future research should develop longitudinal approaches for evaluating causality and the evolution of correlations. Lastly, as suggestions for future research, other researchers should investigate whether these relationships hold true in samples selected through a randomized procedure, and future studies should collect information about the professional sectors of the sample to obtain more rigorous and precise data. Additionally, they should aim to acquire a more extensive sample, particularly in terms of educational background, as our sample suffered from an overrepresentation of individuals with a university education.

In conclusion, social support may promote relaxation processes within the workplace that lead to an increase in job satisfaction and, additionally, a reduction in emotional exhaustion. These considerations hold value and contribute to the exploration of whether there are differences between relaxation derived from social support and other forms of physical-oriented relaxation. Future research should delve deeper into this question.

**Author Contributions:** Conceptualization, S.F.-S. and P.G.; methodology, G.T. and S.F.-S.; software, A.I.H.G.; validation, S.F.-S., G.T. and P.G.; formal analysis, A.I.H.G. and G.T.; investigation, P.G.; resources, P.G.; data curation, P.G.; writing—original draft preparation, S.F.-S.; writing—review and editing, G.T.; visualization, P.G.; supervision, G.T. All authors have read and agreed to the published version of the manuscript.

**Funding:** This research received no external funding.

**Institutional Review Board Statement:** The study was conducted according to the guidelines of the Declaration of Helsinki, and approved by the Institutional Review Board (or Ethics Committee) of Comité de Bioetica de la Universidad Nacional de Educación a distancia (protocol code n/a and date of approval is 17 July 2020).

**Informed Consent Statement:** Informed consent was obtained from all subjects involved in this study.

**Data Availability Statement:** The data presented in this study are available on request from the corresponding author. The data are not publicly available due to privacy.

**Conflicts of Interest:** The authors declare no conflict of interest.

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
