# Peer review of "Social Support and Its Impact on Job Satisfaction and Emotional Exhaustion"

_ejihpe, doi:10.3390/ejihpe13120195_

Round 1

Reviewer 1 Report

Comments and Suggestions for Authors

Dear authors of “Social Support and its Impact over Job Satisfaction and Emotional Exhaustion”. Thank you for focusing on this important area and provide this study. In particular, research on social support at work has lost attention in recent studies and deserves to be focused. I have carefully read the manuscript and have some remarks that in part are somewhat critical. However, I hope they may help you to improve the manuscript.

1.        The abstract is plain, but I did not come to terms with the statements about mediation and moderation – they are different indeed and this should be clarified.

2.       The introduction is too long, wordy, and difficult to read. I would suggest a reduction to 25 % of the length and then clearly point out the central concepts, their relationships, and the importance of the topic, followed by a clear aim and reach question(s). It would also be a strength if you could briefly state how the study contribute with new insights to this research field. In the current version, aims of the study pops up several places in the text, which is confusing.

3.       The theory review –

a.       Line 130 – 132, the sentence “On the other hand, some scholars have included social support as a mediator variable between stress and emotional exhaustion [21], however, it has been directly related to emotional exhaustion” is confusing and needs rewriting.

b.       Line 137: the style of the reference “; Green, 2000)” is not in concert with the style applied in the manuscript.

c.        Line 141: the phrase “Since this is an important…” is unclear, you better repeat the concept focused here.

d.       Line 153: The argument “However, the paths in which this happens remain unclear” seems to be to strong, lots are known and the concepts is well established today, ref. the latest work by Schaufeli et al.

e.       Lines 168-178: this paragraph needs rewriting to focus only on the four dimensions of recovery.

f.        The hypotheses should rest on better arguments, and the relationship between the hypotheses and their theoretical foundation would be easier to see if they were presented one by one in relation to the reviewed theory that support each individual hypothesis. I would strongly recommend a rewrite of the review- and hypothesis chapter.

4.       Method

a.       While the sample is described, it is unclear which types of organisations that participated.

b.       Line 226: The sentence “This study complies with all ethical guidelines. “ does not convince anybody about the ethics. Please be clear in this statement about codes or permissions or regulations that were followed.

5.       Results

a.       Line 293: the term “psychological distance” is not found elsewhere in the results. Should it be detachment?

b.       Lines 315 – 317: The sentence “This section may be divided by subheadings. It should provide a concise and precise description of the experimental results, their interpretation, as well as the experimental conclusions that can be drawn.” Seems to be a comment and is not part of the results. Remove.

6.       Discussion

a.       The discussion is short and might be elaborated by discussing more in detail the hypotheses that failed to be supported. In particular, the different factors constituting the ‘recovery’ concept needs elaboration.

b.       There should be a paragraph discussing the rather serious limitation of the study.

In conclusion, the findings seems to be a valid contribution, but the framing, the foundation of the hypotheses along with the theory review, parts of the methods and the discussion need to be reworked. I wish you good luck with this work and hope my comments may contribute to an improvement of the manuscript. 

Comments on the Quality of English Language

Moderate editing of English language required

Author Response

Dear reviewer.

Thank you very much for your time and dedication. We have followed your instructions and we are convinced that they have substantially contributed to improving our article.

In the attached file, we detail the measures taken.

Reviewer 2 Report

Comments and Suggestions for Authors

Dear Authors,

I wanted to express my sincere appreciation for the valuable research you conducted on the impact of social support on job satisfaction and emotional exhaustion. Your work in this area is truly commendable, and it contributes significantly to our understanding of the dynamics within organizations and the well-being of employees. Your findings, particularly the identification of the direct and inverse relationships between social support and job satisfaction and emotional exhaustion, respectively, provide essential insights for both scholars and practitioners. The inclusion of the relaxation factor as a significant moderator further enriches our understanding of the complexities involved in this domain. 

The study is guided by the principles of Positive Psychology and views sustainable organizations through the lens of four fundamental factors: the individual, the group, the organizational, and the inter-organizational. In this context, social aspects of work, such as support, and individual aspects, like recovery, satisfaction, and emotional exhaustion, are vital for creating harmonious work environments. The research acknowledges the active role that individuals play in harmonizing work environments. The harmony among individuals is viewed as a critical factor contributing to the viability and sustainability of these environments. The study recognizes that social relationships in the workplace are internalized and have a significant impact on individuals' aspirations, motivations, and values. Therefore, it emphasizes the importance of analyzing and considering these relationships in-depth. Moreover, the main text suggests that the study's findings have implications for understanding workplace dynamics and improving employee well-being. It also acknowledges limitations and provides recommendations for future research in this area.

My general comments for consideration by the authors:

1.     I have noticed that the affiliations provided in the paper may not fully align with the guidelines and requirements of our esteemed journal. To ensure that your paper complies with the journal's standards and to facilitate a smoother editorial process.

2.     Specifically, I recommend including more detailed main results in the abstract of your paper. While the current abstract provides a general overview of your research objectives and findings, expanding on the key results would greatly improve the abstract's completeness and allow readers to quickly grasp the core contributions of your study.

3.     To enhance the paper's accessibility and readability, you might want to consider shortening the introduction and theoretical framework sections. While the background information is important, it may be beneficial to present it in a more concise manner, ensuring that the reader can quickly grasp the essence of your research. 

4.     I recommend presenting your hypotheses in a table format which provide a structured and concise way to present information. Organizing your hypotheses in this manner would make it much easier for readers to quickly identify and understand your hypotheses.

5.     I appreciate the effort you've put into the text with the numerical data, but I believe that presenting this information in a table format would greatly enhance the readability of your paper.  Specifically, I kindly request that you consider providing a table 1 summarizing the participants' sociodemographic characteristics.

6.     I kindly request that you provide information regarding missing data and the completeness of data collection in your study. This information can significantly contribute to the robustness and reliability of your findings. It would be beneficial to include information about the steps taken to ensure data collection completeness. This could involve descriptions of data collection methods, quality control procedures, and any efforts made to minimize missing data during the data collection phase.

7.     I have a suggestion that may further enhance the clarity of your findings.  I recommend including tables to present the main results, specifically the direct and indirect effects of social support on job satisfaction. Tables provide a structured and visually appealing way to present complex data, making it easier for readers to quickly grasp the key findings

8.     To further enhance the quality and transparency of your study, I recommend the inclusion of a dedicated "5.3. Study Strengths and Limitations" section within your paper. Recognizing both strengths and limitations provides a balanced perspective on the research. It allows you to highlight what was done well and areas where improvements could be made. Propose concrete ways in which these limitations could be addressed or mitigated in future research. This demonstrates your commitment to the quality and rigor of research in this subject area. Suggest areas for future research that your study has illuminated. Point out questions that remain unanswered or emerging trends that deserve further investigation.

I wanted to extend my warm congratulations to you for the meticulous and thorough presentation of the research tools employed in your paper. Your specific descriptions of the Recovery Experience Questionnaire, Brief Index of Affective Job Satisfaction, and Maslach Burnout Inventory are highly commendable. Your attention to detail greatly enhances the transparency and reproducibility of your research. Additionally, I want to applaud you for providing Cronbach's alpha values to assess test reliability. This is an essential aspect of research that helps ensure the robustness and validity of the measurements used in your study.

Best, reviewer.

Author Response

Dear Reviewer,

Thank you very much for your valuable feedback. Your assessment has greatly aided us in enhancing our manuscript and achieving greater academic and scientific quality. We have taken your suggestions into account and have implemented them. In the attached pdf, you can check the response to your comments.

Kind Regards,

Authors.

Reviewer 3 Report

Comments and Suggestions for Authors

I thank the authors for their work and the editor for the opportunity to review this article. The manuscript's theme is very interesting. This research aims to understand the impact of social support on job satisfaction and emotional exhaustion. 

The introduction focuses entirely on sustainability, immediately making it a central construct in the work. However, the research does not cover this concept, at least directly. We are reading a completely different manuscript when we leave the abstract and move on to the introduction. In addition to focusing on sustainability, it focuses on recovery, which are not the central constructs in this investigation. This section should focus on the main concepts, their relationship, and mainly the relevance of the study and its innovative contribution. There is no reference to these last two points.

In the theoretical Framework, the concepts are merely defined; they do not have the necessary depth, and there is no articulation between them. The hypotheses are not adequately explained or substantiated. Concepts and relationships emerge that were not even mentioned throughout the manuscript.

Regarding the method, we do not have the statistical procedures that allow us to replicate the study carried out. The authors should invest in this subsection and not just mention that "All the analysis was conducted using IBM SPSS v.24. For testing mediation models, we used Model 6 for the PROCESS macro, developed by Hayes". This is poor for understanding the depth of the analysis carried out, as well as replicating the study. We have a sample of 496 participants but we know very little about this sample. For example, in which sectors do you work? For the authors, what are qualified employees and non-qualified employees?

The results section also needs greater description and depth. The authors analyze ten research hypotheses in less than two pages. For example, why don't you use simple and multiple linear regressions beforehand? We do not have descriptive statistics for the variables under study. Does gender affect the variables? Affect the relationship between them? Is it the age? or education? Or the experience? The study has many concepts and many hypotheses, but it is underdeveloped. It needs to be understood what the study adds to the existing literature.

The question regarding the innovation of the study increases when the authors in the discussion section state  "The main objective of this research was to explore and confirm the previous research findings between social support, job satisfaction and emotional exhaustion in a general worker sample". If previous studies had already revealed what is new about this study?

The manuscript does not present a conclusion or future directions.

The study has a good sample, and the rationale is understood, but its presentation is very incipient. The authors must make more effort to deepen theory and data analysis.

Author Response

Dear reviewer,
We sincerely appreciate the time you have devoted to assisting us in enhancing our manuscript. We have worked diligently to incorporate your comments and suggestions. We have made every effort to enhance the theoretical framework, statistical presentation, discussion, and recommendations.
We can confidently state that, after implementing your contributions, our research has gained in rigor and comprehensibility.

You can check our comments to your suggestions in the attached pdf. 

Kind regards,

Authors.

Round 2

Reviewer 3 Report

Comments and Suggestions for Authors

Firstly, congratulations to the authors on the improvement of manuscripts. But many improvements are not clear yet.

The hypotheses are shown in a table, and I don’t understand why. The relationship between the hypotheses and their theoretical foundation would be easier to see if they were presented individually about the reviewed theory that supports each hypothesis. I would strongly recommend a rewrite of this. That is, the hypotheses can be shown across the text and not in a table.

Table 2 does not have the complete information that the authors promise in line 273. And it remains unclear which types of organizations participated.

In the results section, we have another table 2. This table has the same errors as in first version, for example, it doesn’t have a mean to variables in study. The analysis presented does not evolve in relation to the analysis of the first version. I continue with the same questions.

The discussion requires improved argumentation. Remains poor.

I advise authors to use track changes in the next revision, it would be much simpler to revise.

Author Response

Dear reviewer.

Thank you very much. Your valuable contributions have greatly helped in the advancement of our manuscript. We hope that with this second review, it will be suitable for publication
